# Impact of Nitrogen Fertilizer Levels on Metabolite Profiling of the *Lycium barbarum* L. Fruit

**DOI:** 10.3390/molecules24213879

**Published:** 2019-10-28

**Authors:** Zhigang Shi, Feng Wei, Ru Wan, Yunxiang Li, Yajun Wang, Wei An, Ken Qin, Guoli Dai, Youlong Cao, Jiayue Feng

**Affiliations:** 1Wolfberry Engineering Research Institute, Ningxia Academy of Agriculture and Forestry Sciences, National Wolfberry Engineering Research Center, Yinchuan 750002, China; wanru2008@163.com (R.W.); lyxyb@163.com (Y.L.); yajun817@163.com (Y.W.); 13037986722@163.com (W.A.); qinken7@163.com (K.Q.); dgl2006swfc@163.com (G.D.); youlongchk@163.com (Y.C.); 2College of Horticulture, Northwest A & F University, Yangling, Xianyang 712000, China; nxnkwf2016@163.com (F.W.); fengjy19151@nwsuaf.edu.cn (J.F.); 3Ningxia State Farm A & F Technology Central, Yinchuan, Ningxia 750002, China

**Keywords:** *Lycium barbarum* L., nitrogen, metabolome, yield, quality

## Abstract

The yield and quality of goji (*Lycium barbarum* L.) fruit are heavily dependent on fertilizer, especially the availability of nitrogen, phosphorus, and potassium (N, P, and K, respectively). In this study, we performed a metabolomic analysis of the response of goji berry to nitrogen fertilizer levels using an Ultra Performance Liquid Chromatography–Electrospray Ionization–Tandem Mass Spectrometry (UPLC-ESI-MS/MS) method. There was no significant difference in the fruit yield or the commodity grade between N0 (42.5 g/plant), N1 (85 g/plant), and N2 (127.5 g/plant). The primary nutrients of the goji berry changed with an increasing nitrogen fertilization. Comparative metabolomic profiling of three nitrogen levels resulted in the identification of 612 metabolites, including amino acids, flavonoids, carbohydrates, organic acids, and lipids/alcohols, among others, of which 53 metabolites (lipids, fatty acids, organic acids, and phenolamides) demonstrated significant changes. These results provide new insights into the molecular mechanisms of the relationship between yield and quality of goji berry and nitrogen fertilizer.

## 1. Introduction

*Lycium barbarum* L., also known as goji, is a perennial deciduous shrub planted in large areas in the west of China. It prefers to grow in sunny; cold weather; and moist, fertile soil [1]. Goji berry is also an excellent medicinal herb and healthy food. The fruit is rich in vitamins, essential amino acids, and linoleic acid [2]. These nutrients can promote blood flow and prevent arteriosclerosis. Long-term consumption of goji berry can also help to prevent fat accumulation in the liver, improve cell metabolism, and ensure regular body metabolism, thus playing an essential role in slowing down aging of the body [3]. Furthermore, goji berry also has effects on the reduction of inflammation and fever, the elimination of phlegm, liver protection, diabetes prevention, cough relief, moistening of the lungs, and mucus removal [1,4,5]. Recently, the beneficial effects of goji berry have been gradually recognized by people, and the fruit is rapidly becoming an indispensable food supplement. Previous studies have shown that total phenolics, total flavonoid, and total polysaccharide contents of goji fruits from 13 different regions in China ranged from 6.899–8.253 mg gallic acid equivalents/g dry weight (DW), 3.177–6.144 mg rutin equivalents/g DW, and 23.62–42.45 mg/g DW, indicating that the goji berry from Guyuan of Ningxia was most similar to Daodi herbs [6]. Liu et al. reported a comparative analysis of carotenoid accumulation in red fruit of *Lycium barbarum* L. and black fruit of *Lycium ruthenicum* L., finding more extensive contents of carotenoids in red fruit than in black fruit [7].

Nitrogen fertilizer is an essential nutrient that has an important impact on crop growth, yield, and product quality [8]. The proper application of nitrogen fertilizer can increase the soil’s nitrogen content for better plant growth. Several studies investigated the absorption, assimilation, and effects of nitrogen fertilizer on plant development [9]. Nitrogen is mainly absorbed in the form of nitrates and ammonium salts, but may also be absorbed in organic form (e.g., urea or amino acids). The assimilation of inorganic N and organic N requires a series of strictly regulated enzyme catalysis steps [10,11,12]. The growth and development, yield, and quality of goji are all quantitatively limited by nitrogen demand, where either too much or too little nitrogen can inhibit the growth and development of goji. Previous studies showed that about 20% of the nitrogen absorbed during the growth and development of goji comes from fertilizers, with 80% coming from the soil [13]. In two-year-old goji, nitrogen uptake from the fast-growing period to the fruit-harvesting period (from early May to mid-August) continuously increased during the annual growth cycle and in early May to late June. The nitrogen content of the fertilizer absorbed by the plant was higher than that of the total nitrogen, indicating that these two periods were the peak period of nitrogen absorption and the key period for nitrogen fertilizer application [14,15,16]. It was also found that the greatest effect of fertilizers on the growth of goji shoots in spring time was observed in response to nitrogen fertilizer, followed by phosphate fertilizer and, lastly, potash fertilizer [17]. Most of the studies on goji berry mainly focused on the effects of goji berry on human health, as well as the antioxidant activity of goji berry through phytochemical analysis [15,18,19,20,21]. The impact of fertilizer on the metabolites of goji berry has not yet been reported.

Although many reports have identified the pharmacological properties [22], biological activity [2,23], and nutrient benefits of goji berry [5,6,7,24], the relationship between fertilization and the goji berry metabolome is still under-reported. In this study, we utilized an integrated UPLC-ESI-MS/MS detection system to study the effects of different nitrogen fertilizations on the yield and metabolites of goji berry and provided a theoretical model for the regulation of goji berries’ nutrient compounds by fertilization.

## 2. Results

### 2.1. Yield and Commodity Grade of Goji Dry Fruits under Different Nitrogen Fertilizations

The yield and commodity grade of goji fruits under different nitrogen fertilizations were investigated (Figure 1). The yield of goji dry fruit weight (Figure 1A) for the N1 (239.4 ± 4.72a kg/ha) and N2 (231.2 ± 5.78a kg/ha) groups were slightly higher than that of N0 (223.1 ± 3.48a kg/ha), but the difference was not significant (*p* < 0.05). We also analyzed the effects of different nitrogen fertilizations on the commodity grade of goji fruits (Figure 1B). As shown in Figure 1B, we randomly sampled 1 kg fruits of each treatment and repeated three times, which found the commodity grade of the goji fruits denoted 5A, 4A, and 3A was significantly different (*p* < 0.05); the 5A fruits accounted for the highest proportion, with N0, N1, and N2 values of 52.3%, 63.1%, and 62.1%, respectively. The 3A fruits accounted for the lowest ratio, where the N0, N1, and N2 values were 16%, 11%, and 9%, respectively.

### 2.2. Nutritional Contents of Goji Fruits under Different Nitrogen Fertilizations

The contents of polysaccharides, amino acids, total flavonoids, and betaines in goji fruits were analyzed. Seventeen amino acids were detected in the fruits of the 3 treatments, among which 6 essential amino and 11 non-essential amino acids were observed (Figure 2A). The contents of aspartic acid, serine, phenylalanine, and lysine were significantly different between N0, N1, and N2. The highest total contents of essential amino acids were observed under N2 (0.774 g/kg Fresh Weight (FW)) and the lowest under N0 (0.633g/kg FW). The highest total contents of non-essential amino acids were observed under N1 (16.811 g/kg FW) and lowest under N0 (13.695 g/kg FW). The contents of total flavonoids were significantly different between N0, N1, and N2 (Figure 2B), and the total flavonoids increased with the nitrogen fertilization. The contents of polysaccharides (Figure 2C) were significantly different between N0, N1, and N2, with the highest content observed in N0 (13.26g/100g FW) and the lowest content observed in N2 (6.10 g/100g FW). The contents of betaine in N0, N1, and N2 were not significantly different (Figure 2D).

### 2.3. Metabolic Profiling of Goji Fruits under Different Nitrogen Fertilizations

The contents of polysaccharides and the total flavonoids of the goji fruits changed with the nitrogen fertilization in N0, N1, and N2 (Figure 2B,C). The metabolic profiles of the polysaccharides and flavonoids were analyzed for each comparison group using the fold change and variable importance in project (VIP) values. The criteria for analysis included a fold change value of ≥2 or ≤0.5 and a VIP value of ≥1. The results showed that the total contents of the flavones increased with the increase in nitrogen fertilization—mainly due to 3′,4′,5′-tricetin *O*-rutinoside, chrysoeriol *O*-glucuronic acid-*O*-hexoside, tricin 7-*O*-hexosyl-*O*-hexoside, and tricin 5-*O*-hexoside—which were significantly affected by the nitrogen fertilization (Figure 3A). The contents of polysaccharides decreased when nitrogen increased, possibly due to the contents of sucrose and arabinose decreasing (Figure 3B).

### 2.4. Metabolite of Goji Fruits under Different Nitrogen Fertilizations, Functional Annotation, and Enrichment Analysis of KEGG

The secondary metabolites of the goji fruits from three treatments were investigated using a UPLC-ESI-MS/MS method, and the results were compared with databases. The results showed that 612 metabolites were identified in N0, N1, and N2, including 28 amino acids, 55 amino acid derivatives, 13 benzoic acid derivatives, 3 pyridine derivatives, 7 alcohols and polyols, 6 cholines, 10 catechin derivatives, 20 phenolamides, 53 nucleotide and its derivatives, 13 anthocyanins, 42 flavones, 2 flavonolignans, 6 isoflavones, 27 flavonols, 24 flavone C-glycosides, 15 flavanones, 21 quinates and their derivatives, 34 hydroxycinnamoyl derivatives, 3 tryptamine derivatives, 6 alkaloids, 20 carbohydrates, 15 vitamins, 2 terpenoids, 4 coumarins, 4 nicotinic acid derivatives, 5 indole derivatives, 63 organic acids, 2 proanthocyanidins, 34 lipids—glycerophospholipids, 17 lipids—glycerolipids, 19 lipids—fatty acids, and 30 others (Appendix A). 

The different metabolites under the influence of nitrogen fertilizer, which interact with each other in plants, forming different pathways and annotating and displaying differential metabolites by the Kyoto Encyclopedia of Genes and Genomes (KEGG) database (Figure 4). The KEGG enrichment classification results indicated that the impact of different nitrogen fertilization on each group mainly involved biosynthesis of secondary metabolites, glutathione metabolism, flavone and flavonol biosynthesis, isoflavonoid biosynthesis, and flavonoid biosynthesis (Appendix A). 

## 3. Discussion

Goji (*Lycium barbarum* L.) is an outstanding food and traditional medicinal plant in China. After several years of development, the goji industry in China still suffers from a lack of varied agronomic practices and varieties that are not correctly matched to their best method of cultivation. Thus, to study the impact of a single fertilizer on the synthesis of active ingredients in goji fruit, improving the integrated technology for water and fertilizer in goji cultivation based on original technology and methods would be helpful.

### 3.1. Effects of Different Nitrogen fertilizations on Goji Fruit Commodity Grade and Yield

In the present study, a comparison of the yield and commercial grade of fruits in different groups showed that there were no significant differences with increasing nitrogen fertilization. Shi et al. found that, according to the regression model of the optimal design, the mathematical investigating the relationship between yield and various fertilizer factors and the first-order coefficient showed a larger effect for potassium, followed by phosphorus and then nitrogen [17]. It was indicated that potassium and phosphorus were more critical for goji fruits yield than nitrogen. There are also reports that the fruit yield of goji is the result of many agronomic characters, such as branch number, branch length, and crown width [8,25,26]. The fruit commodity rate of N0, N1, and N2 was highest in the 5A group and lowest in the 3A group. It can be inferred that the factor affecting the fruit commodity grade was varied, which still need to further study.

### 3.2. Effects of Different Nitrogen Fertilizations on Fruit Nutrient Content of Goji 

The polysaccharides, amino acids, flavonoids, carotenoids, betaines, vitamins, anthocyanins, and other functional ingredients in goji fruit have been shown to enhance human immunity, inhibit the growth of tumor cells, delay aging, resist fatigue, lower blood pressure, protect the liver, protect vision, and provide antioxidant activity [1,3]. In the present study, seventeen kinds of amino acids were detected in the fruits in the three treatments, with the contents of Pro, Ala, Ser, Asp, Thr, Arg, and Glu being relatively high. The contents of aspartic acid, serine, phenylalanine, and lysine were significantly different between N0, N1, and N2. Notably, Pro was the most available amino acid in all three groups—i.e., N0, N1, and N2—which suggests that high-N fertilizer treatment enhanced the relations of the amino acids and accelerated the transformation amino acids into other types [12,27,28]. We also found that the total content of essential amino acids was highest in N2 (0.774 g/kg FW) and lowest in N0 (0.633g/kg FW), while the total content of non-essential amino acids was highest in N1 (16.811 g/kg FW) and lowest in N0 (13.695 g/kg FW). This is similar to previous research results, which showed that in the range of 0–12 kg of pure nitrogen applied per 666.7 m^2^, the content of various essential amino acids in wheat grains increased when the amount of nitrogen applied increased. The relative contents of lysine, threonine, isoleucine, leucine, and phenylalanine increased, for each addition of 1 kg of pure nitrogen [28]. The nitrogen application had an obvious incremental effect on the content of the essential amino acids in protein, and the high nitrogen fertilizer treatment accelerated the accumulation of these nitrogenous amino acids, resulting in increased protein content [29]. In general, nitrogen-containing molecules—such as glutamic acid, glutamine, and asparagine—play an important role in the assimilation, circulation, translocation, and storage of N [27,28,30,31]. 

The contents of the polysaccharides were significantly different in N0, N1, and N2, decreasing with nitrogen fertilization. The polysaccharide observed was composed of six sugars of arabinose, glucose, galactose, mannose, xylose, and rhamnose [8]. We analyzed the response of the polysaccharide constituents to nitrogen fertilization, finding that the contents of sucrose and arabinose significantly decreased with the nitrogen fertilization (Figure 2C and Figure 3B). A higher concentration of polysaccharides was noted in treatments with low N fertilization, suggesting that an increasing rate of N application led to a decrease in the polysaccharide concentration [8,32]. The contents of total flavonoids increased with nitrogen fertilization, mainly due to the flavones 3′,4′,5′-tricetin *O*-rutinoside, chrysoeriol *O*-glucuronic acid-*O*-hexoside, tricin 7-*O*-hexosyl-*O*-hexoside, and tricin 5-*O*-hexoside, which were significantly affected by the nitrogen fertilization (Figure 2B and Figure 3A, and Appendix A). The effects of nitrogen, phosphorus, and potassium on the contents of polysaccharides and flavonoids in three-year-old goji fruit were ranked as K fertilizer having the largest effect, followed by N fertilizer and then P fertilizer [14,19]. Betaine, the main bioactive component of *L. barbarum*, was reported to possess osmotic regulatory properties in both plants and animals [1]. In the present study, the betaine contents in N0, N1, and N2 were not significantly different. This may have been because all three treatments provided sufficient N fertilizer for plant growth. Under poor growing conditions, plants require nutrients to maintain life instead of synthesizing the secondary metabolite, betaine [8].

### 3.3. Metabolic Profiling of Goji Fruits under Different Nitrogen fertilizations 

Metabolites are the basis of the phenotype of organisms and help us understand the biological processes and mechanisms more intuitively and efficiently. This study was based on a metabolic analysis of a broadly targeted metabolome technique that detected 612 metabolites over three treatment groups, N1, N2, and N0. Kokotkiewicz et al. report that although *L. barbarum* berries are widely recognized as safe to consume, its position in nightshade family may raise some concern about the possible presence of toxic tropane and steroidal alkaloids, such as α-solanine, α-chaconine, solanidine, l-hyoscyamine, and scopolamine [33,34]. In present study, none of the tropane and steroidal alkaloids were detected in fruits. There are two terpenoids (phytocassane C and cucurbitacin D), and six alkaloids (hordenine, piperidine, isoquinoline, camptothecin, betaine, and trigonelline). In comparison to N0, most of the flavonoid metabolites in the fruits of N1 and N2 were upregulated. Flavonoids are a type of valuable secondary metabolite produced from plants that have been reported to possess various beneficial properties, such as anti-cancer activity, anti-inflammatory activity, antioxidant activity, bone brittleness reduction, and so on. These many different types of pharmacological effects have been made more clear due to the development of synthetic dihydrogen flavonoids of naringenin and pinocembrin, which are then further branched into isoflavones, flavonols, flavanones, anthocyanins, etc. [19,35]. Compared with N2, nine kinds of metabolites were found to significantly increase in the N0 treatment group, mainly phenolamide metabolites. phenolamides constitute a class of abundant secondary metabolites obtained by coupling a phenolic moiety to a polyamine or a deaminated aromatic amino acid, which contributes to cell wall enhancement and direct toxicity toward predators and pathogens as a built-in or induced defense [24,36]. Compared with N0, most of the flavonoid metabolites in the fruits of N2 were upregulated. The difference of KEGG enrichment classification under different nitrogen fertilizations also most relate on biosynthesis of secondary metabolites, glutathione metabolism, flavone and flavonol biosynthesis, isoflavonoid biosynthesis, and flavonoid biosynthesis. 

In conclusion, the nitrogen fertilization was in the range of 42.5–127.5 g/plant, which showed no significant difference in the dried fruit yield and fruit commodity grade of goji ‘0909’. Some main nutrients of the fruit, such as flavonoids, amino acids, and polysaccharides, were significantly affected by nitrogen fertilization. There were 612 metabolites identified in the N0, N1, and N2 fruits, and a total of 53 metabolites were significantly affected by the nitrogen fertilization. 

## 4. Materials and Methods 

### 4.1. Plant Materials

Two years old goji seedlings were provided by the Ningxia Academic Agriculture and Forestry Science, Yinchuan, Ningxia, China. The cultivated variety was named ‘0909’.

### 4.2. Experimental Design and Soil Conditions

A field study was conducted at the experimental farm of Ningxia Academic Agriculture and Forestry Science, Yinchuan, Ningxia Hui Autonomous Region, China. The chemical properties of the soil were: pH, 8.44; electrical conductivity, 0.078 dS/m; total salt, 0.52 g/kg; organic matter, 4.86 g/kg; total N, 0.34 g/kg; total P, 0.62 g/kg; total K, 18.50 g/kg; available N, 19.00 mg/kg; available P, 9.00 mg/kg; available K, 158.00 mg/kg; Ca and Mg were 22.80 and 9.20 g/kg, respectively; Cu, Zn, Mn, Cu, Fe, and Se were 45.00, 364.00, 14.90, 20.10, and 0.08 mg/kg, respectively. Seventy seedlings were randomly assigned to each treatment. The treatments were applied in a randomized complete block design, and each treatment had four replications. The total fertilizer needed for the goji was split into nitrogen-based fertilizer: 60% (14 March), top dressing 20% (15 May), and top dressing 20% (13 August); phosphate-based fertilizer: 40% (14 March), top dressing 30% (15 May), and top dressing 30% (13 August); and potash-base fertilizer: 40%(14 March), top dressing 30% (15 May), and top dressing 30% (13 August). The details of each treatment are shown in Table 1. The nitrogen, phosphorus, and potassium were supplied by urea (N 46%), ammonium diacid phosphate (N 12%, P_2_O_5_ 61%), and potassium sulfate (K_2_O 52%), respectively.

### 4.3. Determination of Primary Nutrients of the Goji Fruit

#### 4.3.1. Determination of Total Flavonoid, Polysaccharide, and Betaine Contents

Each sample of treated fruit weighed about 5.0 g; these were freeze-dried using a vacuum freeze dryer (LGJ-10N-50A, Li Chen Instrument Technology Co., Ltd., Shanghai, China). After cutting the samples, the samples were rapidly frozen using liquid nitrogen and were maintained at 2–8 °C after melting; then, PBS (1 g sample was added 9 mL PBS) was added (pH 7.4). The samples were homogenized by hand or using grinders, until the sample was completely pulped. Then centrifuged (Centrifuge 5424 R, Eppendorf China Co., Ltd., Shanghai, China) for 20 min at 2000–3000 rpm, and then the supernatant was removed. The total flavonoid content of 0.05 g fruits was determined according to the protocol of the plant flavonoid test kit (Shanghai Enzyme-linked Biotechnology Co., Ltd., Shanghai, China). The polysaccharide content of 0.03 g fruits was determined in accordance with the protocol of the plant polysaccharide test kit (Shanghai Enzyme-linked Biotechnology Co., Ltd., Shanghai, China). The betaine content of 0.03 g fruits was determined in accordance with the protocol of the plant betaine test kit (Shanghai Enzyme-linked Biotechnology Co., Ltd., Shanghai, China) [37]. The protocol of kit assay of plant flavone, polysaccharide, and betaine are provided (Supplementary Plant ELISA Kit protocol).

#### 4.3.2. Determination of the Amino Acid Content

The appropriate amount of crushed sample material was weighed and placed in a 15 mL centrifuge tube, followed by 5 mL of 0.02 mol/L hydrochloric acid, which was then swirled and mixed for 5 min. Ultrasonic extraction was carried out for 5 min, and the sample was placed in the dark for 2 h. Then, 0.02 mol/L hydrochloric acid was added at a constant volume of 10 mL. After centrifugation (Centrifuge 5424 R, Eppendorf China Co., Ltd., Shanghai, China) for 10 min, 1 mL of the supernatant was accurately collected, and 1 mL of 6–8% sulfosalicylic acid was added. This was vortexed for 1 min, left in the dark for 1 h, centrifuged (Centrifuge 5424 R, Eppendorf China Co., Ltd., Shanghai, China) for 15 min at 15,000 rpm, and then the supernatant was obtained. The derivative of 250 μL phenylacetonitrile isocyanate and 250 μL triethylamine acetonitrile was obtained after 1 h and then added to 2 mL n-hexane. The liquid was layered, and the lower layer was collected and passed through the film on the machine with the following conditions: mobile phase A—0.1mol/L sodium acetate solution: acetonitrile (97:3) and mobile phase B—acetonitrile:water (8:2). Chromatographic conditions and system applicability: Silica gel bonded with octadecyl silane was used as a filler (4.6 × 25 mm, 5 μm), and the flow rate was 1 mL per minute. The column temperature was 40 °C, the detection wavelength was 254 nm, and the peak separation of each amino acid was greater than 1.0. The elution gradient is shown in Table 2. The mass spectrometry of the amino acids is shown in Appendix A. The sample extracts were analyzed using an LC-MS system (ThermoFisher U3000, Thermo Fisher Scientific China Co., Ltd., Shanghai, China) and a 17 amino acid mixture standard (Sigma-Aldrich, St. Louis, Missouri, USA). Concentrated phenyl isothiocyanate triethylamine acetonitrile (chromatographic pure) sodium acetate reagent was purchased from Sinopharm Chemical Reagent Co., Ltd (Shanghai, China).

### 4.4. Sample Preparation and Extraction

The freeze-dried fruit was crushed using a mixer mill (MM 400, Retsch) with a zirconia bead for 1.5 min at 30 Hz. Samples of powder weighing 100 mg were extracted overnight at 4 °C with 1.0 mL of 70% aqueous methanol. Following centrifugation (Centrifuge 5424 R, Eppendorf China Co., Ltd., Shanghai, China) at 10,000 g for 10 min, the extracts were absorbed (CNWBOND Carbon-GCB SPE Cartridge, 250 mg, 3 mL; ANPEL, Shanghai, China, www.anpel.com.cn/cnw) and filtrated (SCAA-104, 0.22 μm pore size; ANPEL, Shanghai, China, http://www.anpel.com.cn/) before LC-MS analysis.

### 4.5. HPLC Conditions

The sample extracts were analyzed using an LC-ESI-MS/MS system (HPLC, Shim-pack UFLC SHIMADZU CBM30A system, Redwood City, USA www.shimadzu.com.cn/; MS, Applied Biosystems 6500 Q TRAP, www.appliedbiosytem.com.cn/). The analytical conditions were as follows. HPLC column, water ACQUITY UPLC HSSS T3 C18 (1.8 μm, 2.1 μm × 100 mm); solvent system, water (0.04% acetic acid), acetonitrile (0.04% acetic acid); gradient program, 95:5 *v/v* at 0 min, 5:95 *v/v* at 11.0 min, 95:5 *v/v* at 12.0 min, 95:5 *v/v* at 15.0 min; flow rate, 0.40 mL/min; temperature, 40 °C; injection volume, 2 μL. The effluent was alternatively connected to an ESI-triple quadrupole-linear ion trap (Q TRAP)–MS.

### 4.6. ESI-Q TRAP-MS/MS

Mass spectrometry followed the method of Chen et al. [38]. Linear ion trap (LIT) and triple quadrupole (QQQ) scans were acquired on a triple quadrupole–linear ion trap mass spectrometer (Q TRAP), API 6500 Q TRAP LC/MS/MS System, equipped with an ESI Turbo Ion-Spray interface, which was operated in both positive and negative ion mode and controlled via Analyst 1.6 (AB Sciex, Concord, ON, Canada). The ESI source operation parameters were as follows: Ion source, turbo spray; source temperature 500 °C; ion spray voltage (IS) 5500 V; ion source gas I (GSI), gas II (GSII), and curtain gas (CUR) were set at 55, 60, and 25.0 psi, respectively. The collision gas (CAD) was high. Instrument tuning and mass calibration were performed with 10 and 100 µmol/L polypropylene glycol solutions in QQQ and LIT modes, respectively. QQQ scans were performed as multiple reaction monitoring (MRM) experiments, with the collision gas (nitrogen) set to 5 psi. The declustering potential (DP) and the collision energy (CE) for individual MRM transitions were determined with further DP and CE optimization. A specific set of MRM transitions was monitored for each period according to the metabolites eluted within this period.

### 4.7. Qualitative and Quantitative Analyses of Metabolites

Qualitative and quantitative analyses of metabolites followed the methods of Wang [39] and Fraga [40]. Based on the self-built database MWDB (Metware Biotechnology Co., Ltd. Wuhan, China) and the public database of metabolite information, qualitative analyses of the primary and secondary spectral data of mass spectrometry were performed. The analyses removed the isotope signal, a repetitive signal containing K^+^ ions, Na^+^ ions, NH_4_^+^ ions, and a repetitive signal of fragment ions with larger molecular weights. The quantitative analysis of metabolites was performed using MRM analysis of QQQ mass spectrometry. After obtaining the metabolite mass spectrometry data of different samples, peak area integration was performed on the mass spectrum peaks of all substances, and the mass spectrum peaks of the same metabolite in different samples were integrated for correction.

### 4.8. Sample Quality Control Analysis

The high stability of the instrument guaranteed data repeatability and reliability. To check the repeatability of metabolite extraction and detection, overlapping display analyses of total ion current (TIC) maps for mass spectrometry analysis of different quality control samples (QC) were drawn. The QC was prepared by mixing sample extracts and for every 10 test samples, one QC was inserted. The stacking diagram of TIC maps from QC mass spectrometry is shown in Appendix A. The TIC curve of the metabolites showed high overlap, i.e., the retention time and peak intensity were consistent, indicating that the signal stability was good when the mass spectrometer detected the same sample at different times.

### 4.9. Statistical Analysis

Statistical software was performed using Microsoft Office Excel 2013 and SPSS 20.0 (IBM Corporation, Armonk, NY, USA). One-way analysis of variance (ANOVA) and Duncan’s multiple range test to determine the significant difference were also performed. Significance was set at *p* < 0.05. All figures in this article were drawn using Graphpad Prism 7.0 (GraphPad Software, Inc., 7825 Fay Avenue, Suite 230, La Jolla, CA 92,037 USA). Orthogonal signal correction and partial least squares-discriminant analysis (OPLS-DA) was carried out by using R (http://www.r-project.org/) [41].

## Figures and Tables

**Figure 1 molecules-24-03879-f001:**
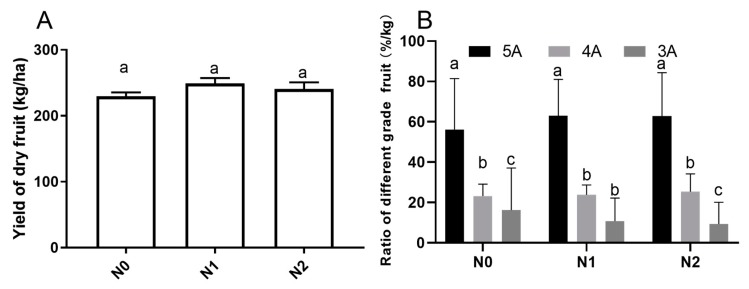
Yield and commodity grade of goji fruits under different Nitrogen fertilizations. Small letters indicate a significant difference (*p* < 0.05), as analyzed by Duncan’s multiple tests. The grading standard of the commodity grade of goji berry is according to the state council, the Ministry of Commerce and Health, GBT18672-2014 and other relevant national standards, as well as the particle size. The number of particles contained in 50 g fruits are: 5A—180 particle/50g; 4A—220 particle/50g; 3A—280 particle/50 g; 2A means inferior fruits, which not shown in the bar graph. (**A**) Yield of goji dry fruit; (**B**) Ratio of different grade goji fruit.

**Figure 2 molecules-24-03879-f002:**
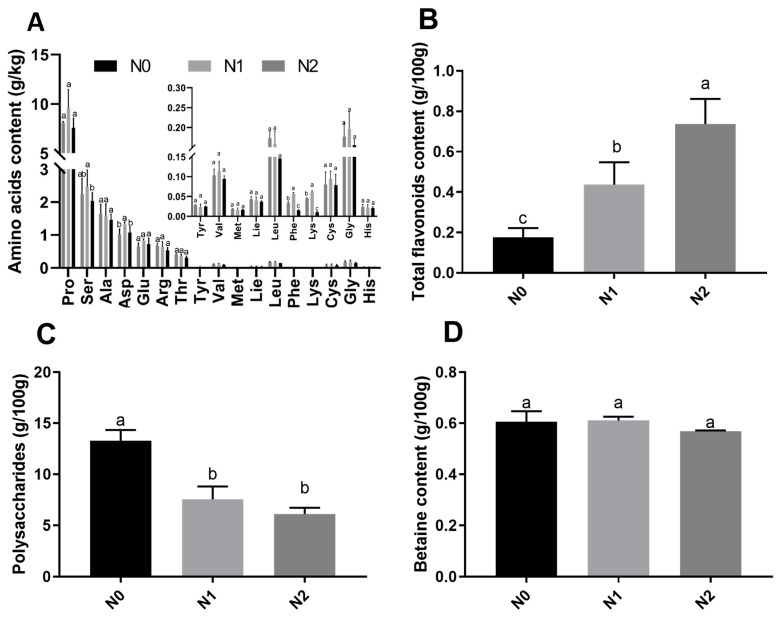
Nutrient contents of goji fruits. The small letter indicates a significant difference (*p* < 0.05) as analyzed by Duncan’s multiple tests. (**A**) Amino acid contents of goji fruits; (**B**) total flavonoid contents of goji fruits; (**C**) polysaccharide contents of goji fruits; and (**D**) betaine contents of goji fruits.

**Figure 3 molecules-24-03879-f003:**
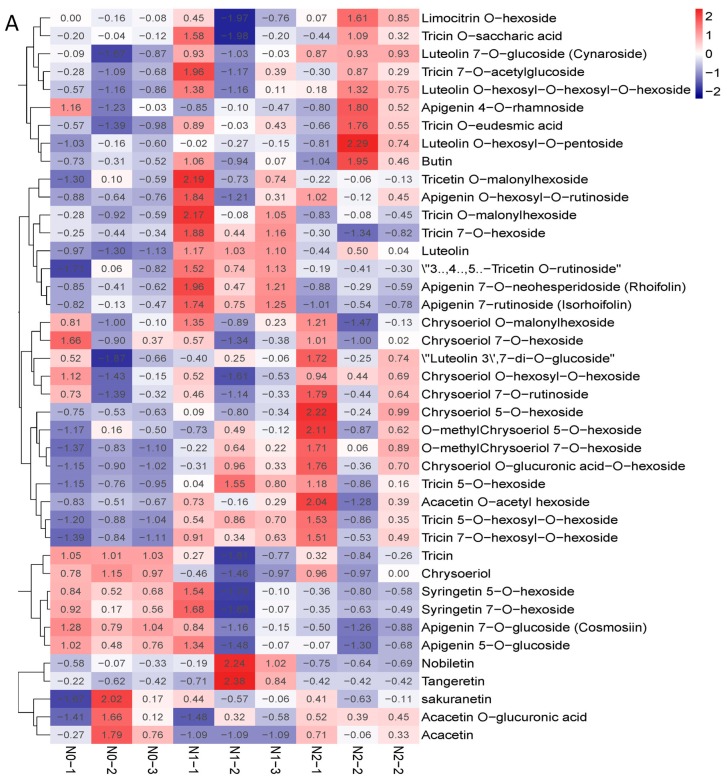
(**A**) Heat map of flavone metabolites of goji berry; (**B**) heat map of polysaccharide of goji berry. There is one column for each sample and one row for each metabolite. The abundance of each metabolite is represented by a bar with a specific color. The upregulated and downregulated metabolites are expressed in different shades of red and blue, respectively. As abundance increases, the color of the bar changes from blue to red. When the abundance value is 0, the color of the bar is white, as shown in the upper right-hand bar.

**Figure 4 molecules-24-03879-f004:**
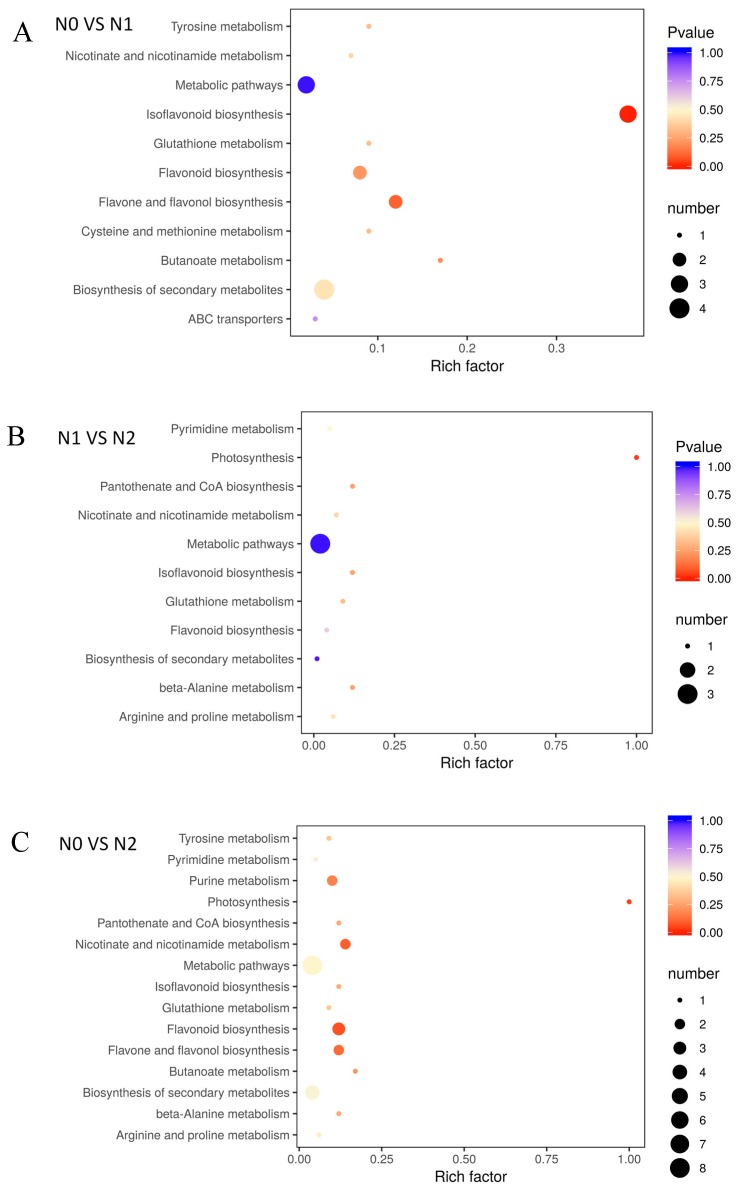
Enrichment analysis of KEGG. (**A**) N0 VS N1; (**B**) N1 VS N2; (**C**) N0 VS N2; The *p*-value represents the degree of enrichment, and the *p*-value is close to 0, indicating that the enrichment is more significant. The size of point represents the number of differential metabolites.

**Table 1 molecules-24-03879-t001:** Experimental treatments.

Treatment	Nitrogen g/plant	Phosphorus g/plant	Potassium g/plant
N0	42.5	65	50
N1	85	65	50
N2	127.5	65	50

**Table 2 molecules-24-03879-t002:** Elution gradient.

Time (min)	Mobile Phase (A) %	Mobile Phase (B) %
0	100	0
14	85	15
29	66	34
30	0	100
37	0	100
38	100	0
45	100	0

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
