# Peer review of "Impact of Nitrogen Fertilizer Levels on Metabolite Profiling of the Lycium barbarum L. Fruit"

_molecules, 2019, doi:10.3390/molecules24213879_

Round 1

Reviewer 1 Report

The manuscript has improved after revision.

Some minor revision is necessary.

Line 65: "The impact of fertilizer on the metabolites of Goji has not yet been reported", after this sentence, the references are not appropriate.

Paragraphs 2.2 and 3.2: the nutrient contents are on dry weight?

Lines 108 and 116: the authors should check if "polysaccharides" and "polysaccharide metabolites" is appropriate. 

Lines 173-175: the same values have been reported in lines 94-96.

Lines 106-107: already commented in the previous paragraph.

Line 207: "medicine sources", please replace the term.

Lines 256-259: at least the principle of analytical determinations and the references should be reported.

Line 273: the authors should indicate the column and the brand.

Author Response

Dear  Reviewer:

Thank you for your comments concerning our manuscript entitled “The Impact of Nitrogen Fertilizer Levels on Metabolite Profiling of the Lycium barbarum L. Fruit” (ID: molecules-600211).Those comments are all valuable and very helpful for revising and improving our paper, as well as the important guiding significance to our researches. We have studied comments carefully and have made correction. The main corrections in the paper and the responds to the comments are as flowing:

1. Line 65: "The impact of fertilizer on the metabolites of Goji has not yet been reported", after this sentence, the references are not appropriate.

Response: Those references are to support “Most of the studies on goji mainly focused on the effects on human health, as well as the antioxidant activity of goji through phytochemical analysis”. So it should cited after this sentence.

2. Paragraphs 2.2 and 3.2: the nutrient contents are on dry weight?

Response: The nutrient contents are on fresh weight.(Fresh fruit samples treated with liquid nitrogen). We have mark it in manuscript.

3. Lines 108 and 116: the authors should check if "polysaccharides" and "polysaccharide metabolites" is appropriate.

Response: We have check it again, it is true the “polysaccharide metabolites” is no appropriate. So we change it to polysaccharides in line 116.

4. Lines 173-175: the same values have been reported in lines 94-96.

Response: The line 173 is report the total content of essential amino acids, the line 175 is report the total content of no-essential amino acids. They are not same values.

5. Lines 106-107: already commented in the previous paragraph.

Response: The contents are on fresh weight(Fresh fruit samples treated with liquid nitrogen).

6. Line 207: "medicine sources", please replace the term.

Response: We check it again, and think it is may be not appropriate. So we remove it from this sentence.

7. Lines 256-259: at least the principle of analytical determinations and the references should be reported.

 Line 273: the authors should indicate the column and the brand.

Response: These problems are supplemented in the manuscript.

Reviewer 2 Report

The paper discusses the use of Goji plants to test effects of nitrogen fertilization and this is followed by an analysis of growth indicators and metabolites. There are parts that need to be rewritten and I have indicated this as such in the document that is attached to this review. 

However, my biggest concern is that the KEGG data analysis really makes little sense to me especially when hits that are being pulled from the data set are linked to human diseases but the study was directly involved with plants and so plant pathways should then be the focus of this. How can you be visualing information linked to the HIV virus,  etc when you are working with Goji Fruit. Please see on the document all the places where I have made comments and indicated the 'metabolites' that are being identified via KEGG analysis that really make absolutely no sense what so ever when considering that this was plant material that was being studied rather than animal tissues.

There were no studies on mammalian systems so I really do not understand why the metabolites that are being identified are linked to mammalian systems. Furthermore, these data are not discussed at all in the discussion session and there is no correlation between materials and methods, results, and discussion linked to this part of the study. 

I am also concerned with some of the graphics as they are very difficult to see and impossible to read. I have made suggestions on how to improve those. 

Author Response

Dear Reviewer:

Thank you for your comments concerning our manuscript entitled “The Impact of Nitrogen Fertilizer Levels on Metabolite Profiling of the Lycium barbarum L. Fruit” (ID: molecules-600211).Those comments are all valuable and very helpful for revising and improving our paper, as well as the important guiding significance to our researches. We have studied comments carefully and have made correction in the paper. Please see the attachment.

Reviewer 3 Report

The paper reported the metabolomic analysis of the response of goji berry (the fruit of Lycium barbarum L.) to nitrogen fertilizer levels using a UPLC-MS method. The study object, goji berry, is a popular medicinal herb and healthy food. The research on the relationship between yield and quality of goji berry and nitrogen fertilizer is of value. I think that it can be accepted for publication after addressing the following issues.

Dicaffeoylspermidine glycosides are major constituents in goji berry. However they were entirely ignored in the research. The high, middle, and low nitrogen fertilizer groups were represented by N2, N0, and N1, respectively. However, the numbering is difficult to be understood. There are also the numbering and sorting questions in the Figures. In the first sentence of Introduction section, the authors stated that “Lycium barbarum L., also known as Goji”. However, goji berry refers specifically to the ripe fruit of Lycium barbarum L., and Lycium barbarum L. refers to the plant. Besides, the name “goji” don’t need to be capitalized except for the first letter. Figures 1A and 1B showed that the commodity rate of goji berry was about 25%, and such a low commodity rate is abnormal. The data, including the yield of goji berry, should be expressed as means ± SD/SEM. There are some English typos.

Lycium barabrum L.” should be “Lycium barbarum L.” (Keywords).

……

Author Response

Thank you for your comments concerning our manuscript entitled “The Impact of Nitrogen Fertilizer Levels on Metabolite Profiling of the Lycium barbarum L. Fruit” (ID: molecules-600211).Those comments are all valuable and very helpful for revising and improving our paper, as well as the important guiding significance to our researches. We have studied comments carefully and have made correction. The main corrections in the paper and the responds to the comments are as flowing:

1. Dicaffeoylspermidine glycosides are major constituents in goji berry. However they were entirely ignored in the research.

Response: In present study, we also found dicaffeoylspermidine glycosides, but it hard to prove it without enough evidences, so we have not discussed it here.

2. The high, middle, and low nitrogen fertilizer groups were represented by N2, N0, and N1, respectively. However, the numbering is difficult to be understood. There are also the numbering and sorting questions in the Figures.

Response:We have revised this problem. The high, middle, and low nitrogen fertilizer groups were represented by N2, N1, and N0.

3. In the first sentence of Introduction section, the authors stated that “Lycium barbarum L., also known as Goji”. However, goji berry refers specifically to the ripe fruit of Lycium barbarum L., and Lycium barbarum L. refers to the plant. Besides, the name “goji” don’t need to be capitalized except for the first letter.

Response:We have checked it carefully and revised it in manuscript.

4. Figures 1A and 1B showed that the commodity rate of goji berry was about 25%, and such a low commodity rate is abnormal. The data, including the yield of goji berry, should be expressed as means ± SD/SEM. There are some English typos.“Lycium barabrum L.” should be “Lycium barbarum L.” (Keywords).

Response:Figure1A is show the yield of goji dry berry. Because of the goji seedlings was two-years-old, and according to our and other previous study, this yield of dry fruit is normal.

Figure2A is show the commodity grade of goji berry, we made a mistake about the bar graph, which use another data of bar graph to express it. We have revised the mistake in manuscript.

Reviewer 4 Report

Manuscript Molecules-600211 is an interesting contribution for understanding the effect of N fertilization on the metabolome of Goji. Nevertheless, it requires some minor corrections for improving its quality:

-          Lines 161-162: Authors should explain why the main factor affecting the fruit commodity grade was the variety, because they have examined only one variety. tHe same is applicable to water supply.

-          Lines 224-228: Please, write “nitrogen fertilization”, not “nitrogen concentration”.

-          Subsection 4.2.1: Which company provided the plant flavonoid test kit and the plant polysaccharide test kit?

Author Response

Dear Reviewer:

Thank you for your comments concerning our manuscript entitled “The Impact of Nitrogen Fertilizer Levels on Metabolite Profiling of the Lycium barbarum L. Fruit” (ID: molecules-600211).Those comments are all valuable and very helpful for revising and improving our paper, as well as the important guiding significance to our researches. We have studied comments carefully and have made correction. The main corrections in the paper and the responds to the comments are as flowing:

1. Lines 161-162: Authors should explain why the main factor affecting the fruit commodity grade was the variety, because they have examined only one variety. The same is applicable to water supply.

Response: we have check the lin161-162 again, and found it is not appropriate. So we have change it in the manuscript.

2. Lines 224-228: Please, write “nitrogen fertilization”, not “nitrogen concentration”.

Response: we have revised it in manuscript.

3. Subsection 4.2.1: Which company provided the plant flavonoid test kit and the plant polysaccharide test kit?

Response: The information of plant ELISA kit protocol are provide in the manuscript and supplementary.

Reviewer 5 Report

Given the importance of goji fruits as a food component, the presented results may be of importance for food chemists and food technologists. Some of the presented results are quite basic (e.g. the influence of nitrogen fertilizer on fruits yield), however, the metabolic profiling studies provide some elements of novelty. My recommendation is to publish the paper after some minor issues are addressed:

Specific remarks:

the supplementary material looks like a raw data. I suggest to provide the figrures and tables with appropriate description (e.g. table and figure headings, chromatogram description with abbreviations, clear descriptions of the column content). For readers convenience, this data should be included in the file. Please also adjust the width of table columns. there have been some concerns regarding the possible presence of tropane and steroid-type alkaloids in goji fruits. This issue was raised by Adams et al. (https://doi.org/10.1002/pca.915) and more recently by Kokotkiewicz et al. (https://doi.org/10.1016/j.foodchem.2016.11.142). Has the metabolomic study revealed the presence of any of these constituents? Please comment on this in the discussion section. Lines 251-256: please provide more specific information concerning the extraction procedure (homogenization time, PBS volume added) Lines 256-260: the protocols can be included in supplementary materials  Figure 4: does it make any sense to include such organismal systems as, for example, "insulin secretion" or "gastric acid secretion" in a research paper on plants? These are irrelevant, in my opinion. Please provide an explanation on why was this data included.

Author Response

Thank you for your comments concerning our manuscript entitled “The Impact of Nitrogen Fertilizer Levels on Metabolite Profiling of the Lycium barbarum L. Fruit” (ID: molecules-600211).Those comments are all valuable and very helpful for revising and improving our paper, as well as the important guiding significance to our researches. We have studied comments carefully and have made correction. The main corrections in the paper and the responds to the comments are as flowing:

1. The supplementary material looks like a raw data. I suggest to provide the figrures and tables with appropriate description (e.g. table and figure headings, chromatogram description with abbreviations, clear descriptions of the column content). For readers convenience, this data should be included in the file. Please also adjust the width of table columns.

Response:We have described the supplementary materials as you suggested.

3. There have been some concerns regarding the possible presence of tropane and steroid-type alkaloids in goji fruits. This issue was raised by Adams et al. (https://doi.org/10.1002/pca.915) and more recently by Kokotkiewicz et al. (https://doi.org/10.1016/j.foodchem.2016.11.142). Has the metabolomic study revealed the presence of any of these constituents? Please comment on this in the discussion section.

Response:Thank you for your suggestion, we have not paid attention to this informations before. We reanalyze the metabolome data and  comment  it in conjunction with your suggested referencesin discusstion section.

3. Lines 251-256: please provide more specific information concerning the extraction procedure (homogenization time, PBS volume added)

Response: 1g sample was added 9ml PBS, and the time of homogenization is until it is completely pulped. The protocol of plant ZLIZA kit was provide in supplementray.

4. Lines 256-260: the protocols can be included in supplementary materials

Response: we have provide the protocols in manuscript and supplementary materials.

5. Figure 4: does it make any sense to include such organismal systems as, for example, "insulin secretion" or "gastric acid secretion" in a research paper on plants? These are irrelevant, in my opinion. Please provide an explanation on why was this data included

Response: Because of there are some metabolites probably have relation with organismal systems. But, we check the figure 4 again, and also felt it is not appropriate. So we have revised the figure 4 in manuscript, just include some paths about plant.
